# Perspective on Immunoglobulin N-Glycosylation Status in Follicular Lymphoma: Uncovering BCR-Dependent and Independent Mechanisms Driving Subclonal Evolution

**DOI:** 10.3390/cancers17071219

**Published:** 2025-04-04

**Authors:** Gloria Pokuaa Manu, Mariette Odabashian, Sergey Krysov

**Affiliations:** 1Barts Cancer Institute, Barts and The London School of Medicine and Dentistry, Queen Mary University of London, London EC1M 6BQ, UK; g.p.manu@hss21.qmul.ac.uk (G.P.M.); marietteodabashian@hotmail.co.uk (M.O.); 2West African Centre for Cell Biology of Infectious Pathogens, Department of Biochemistry Cell and Molecular Biology, University of Ghana, Legon, Accra P.O. Box LG 54, Ghana

**Keywords:** follicular lymphoma, clonal evolution, immunoglobulin variable region, somatic hypermutation, N-Glycosylation, lectins

## Abstract

FL is a complex, incurable disease characterized by the acquired N-glycosylation containing sugar-like molecules (N-glycans) in the immunoglobulin variable regions, which evidently supports FL development and survival. Some rare N-glycan-negative FL subclones survive briefly, suggesting alternative survival mechanisms. This perspective explores how analyzing FL cells’ genetic activity can uncover N-glycan-dependent and independent survival pathways, aiming to advance our understanding of FL pathobiology and facilitating the development of improved therapeutic strategies.

## 1. Introduction

Follicular lymphoma (FL) is a form of non-Hodgkin lymphoma (NHL) with the distinct follicle-like structures within lymph nodes (LNs), frequent bone marrow (BM) involvement, widespread lymphadenopathy, and occasional splenomegaly [1]. The clinical course of the disease varies from an indolent form to a more aggressive disease [2]. Accounting for about 30% of all lymphoma cases and showing a survival rate of ~80% > 10 years, its diagnosis is often made around 60 years old, with most patients presenting at an advanced stage (stage III or IV) of the disease, at which point the disease is incurable [3]. FL treatment strategy varies depending on the disease stage and the severity of symptoms. Options include a watch-and-wait approach for indolent cases, radiation for early-stage disease, chemoimmunotherapy (CHOP) for advanced-stage disease, or combinations of these treatments. Targeted therapies, the rituximab maintenance (R) or the R-CHOP combination, and stem cell transplants can also be used. Treatment aims to control FL, achieving and prolonging remission and improving patients’ quality of life [4]. Although the initial drug response rate is high, its resistance to available drugs is an issue. Also, over 28% of patients experience transformation to a treatment-resistant aggressive disease such as transformed FL (tFL) or diffuse large B-cell lymphoma (DLBCL), both of which have a poor prognosis and treatment outcomes [5,6,7].

FL progression and clonal evolution are influenced by genetic and epigenetic drivers, with CREBBP, KMT2D and EZH2 being amongst the most frequent [8,9]. Complex tumor microenvironmental (TME) interactions are also influential [10]. Importantly, ongoing SHM introduces N-gly motifs, which are asparagine residues in the canonical consensus sequence N-X-S/T (where X can represent any amino acid except Proline) [11], into the immunoglobulin variable (IgV) regions. These motifs facilitate the attachment of unusual high-mannose type N glycans, specifically oligomannose structures (M5–M9), while bypassing the subsequent glycosylation modifications typically associated with ‘mature’ human proteins [12,13,14,15]. The processing of glycan seems to be always halted at the high-mannose stage in FL clones [13]. In FL IgVs, 70–80% of acquired N-gly motifs are found in the complementarity determining region (CDR) and 20–30% are located in the framework region (FR). FL IgV glycosylation and the role of CDR-specific steric hindrance in the ability of α-mannosidases to access the proximal oligomannoses in the processing of glycans on IgV, which results in mannosylated FL IgV, was recently discussed elsewhere [12,13]. FL IgV N-glycosylation requires for the antigen-independent B-cell receptor (BCR) activation through interaction with mannose-binding lectins, with DC-SIGN and the mannose receptor being prime candidates and essential for the survival and proliferation of FL cells [15,16]. A recent next-generation sequencing (NGS) analysis of the immunoglobulin gene heavy chain variable regions (*IGHV*) in a series of longitudinal samples confirmed that N-gly motifs are consistently maintained throughout the progression of the disease and are seemingly independent of intratumoral genetic variability [17]. However, rare N-gly motif-negative subclones have also been identified, which appear to be generally short-lived and typically absent in subsequent samples, although a small number exhibit transient proliferative capabilities [14,17,18,19]. Moreover, the identification of short-lived N-gly-positive non-proliferating subclones, among other possibilities, may reflect structural changes in the IgV that hinder the interaction of IgV mannose with receptors that specifically recognize mannose residues within the TME. Among other explanations, this could be due to the N-gly motif being masked from TME interaction, resulting in the absence of supportive BCR signaling in these subclones.

The presence of FL subclones when the change in N-gly status influences subclonal behavior provides an opportunity for characterizing disease-driving transcription signatures. Here, we examine recent observations suggesting that the IgV N-gly status can help distinguish these transcription signatures in FL subclones. The knowledge of these mechanisms could be leveraged in novel therapeutic approaches that aim to manage FL more effectively.

## 2. Follicular Lymphoma Pathogenesis and Clonal Evolution

The pathogenesis of FL arguably starts with the in situ follicular neoplasia (ISFN) of precursor B cells in LN germinal centers [20]. While the exact genetic changes are still under investigation, the t (14; 18) (q32; q21) translocation, found in 90% of FL cases, is known to place the *BCL2* oncoprotein gene under the control of the *IGVH* gene enhancer, making cells resistant to apoptosis [21,22]. However, it is also evident that this translocation alone does not directly cause FL, as shown in studies in animals and healthy individuals carrying t (14; 18) B cells [23]. This implies a step-wise process of accumulation of several transforming events. The observation of recurrent mutations and the importance of cell crosstalk within the tumor microenvironment suggests that FL pathogenesis involves genetic and epigenetic alterations, as well as interactions with the microenvironment [8,9,10]. An analysis of the EPIC (500,000 individuals) and CPS (120,000 individuals) cohorts highlighted that the *BCL2* gene rearrangement caused by the t (14; 18) translocation, together with the mutation of the *CREBBP* gene, are strongly associated with the FL founder population [9]. Other common mutations in the *KMT2D, EZH2, TNFRSF14* genes and subclonal mutations in the *RRAGC, MEF2B*, and *FOXO1* genes have also been associated with lymphomagenesis [8,9,10].

FL pathogenesis involves multiple steps, with immunoglobulins playing a key role. Although recently challenged [24], it is still widely accepted that *VDJ* recombination on one *Ig* gene allele ensures that each B cell produces one antibody, while employing allelic exclusion of the second *Ig* gene allele. The allelic exclusion prevents usage of the second *Ig* gene allele [25]. The t (14; 18) translocation, thought to result from aberrant RAG1/2 activity during early B cell development, disrupts the functional immunoglobulin gene, seemingly preventing immunoglobulin heavy chain expression, which is crucial for BCR assembly and B cell survival. Surprisingly, affected B cells undergo a full *VDJ* recombination on the second gene allele, enabling BCR expression and allowing pro-survival signals to be received from the tumor microenvironment.

Clonal evolution, a process in which tumor cells undergo genetic and phenotypic changes, is very common with FL cases [8,11,17,18,19,26]. Subclonal evolution in FL contributes to tumor heterogeneity, drug resistance, disease progression and transformation [27]. Importantly, ongoing SHM in FL allows an analysis of subclonal evolution through the characterization of mutations in the *IgV* (*VDJ/VJ)* genes [17,26]. This SHM phenomenon results in a diverse sub-population of FL cells within a single tumor [17]. Such clonal variability is a hallmark of FL that enables certain subclones to gain a selective advantage under various selective pressures, such as in immune surveillance and treatments.

*IgV* sequencing, which characterizes the composition of *IgV*s at high resolution, has been instrumental in tracing the sub-clonal evolution in individual and longitudinal FL samples, revealing the history of distinct subclones and tracking their origins from a common progenitor cell [8,17]. The subclones that develop advantageous traits may proliferate, leading to the dominance of specific clones, while less fit subclones gradually decline [17]. This selective expansion shapes the overall disease course and can lead to the emergence of treatment-resistant subclones.

## 3. Role of N-Glycosylation Motifs in Follicular Lymphoma Clonal Evolution

Immunoglobulins (Igs) are glycoproteins produced by B cells that are involved in adaptive immunity. Normally, Igs undergo N-gly at motifs found in the heavy-chain constant regions or in some IgV germline sites that have a naturally occurring N-gly motif, including V1-08 and V4-34 [28]. N-gly is an important modification for Igs, affecting their stability, folding, and functional capabilities [29]. Moreover, the precise glycosylation patterns are important in modulating the immune response [30]. The Ig variable regions in FL acquire de novo specific N-glycosylation amino acid motifs, N-X-S/T [28,31]. About 90% of IgM+ and 73% of IgG+ cases of FL patients have acquired N-gly motifs in their Ig that have been introduced by SHM in the *IgV* genes and are usually present in the CDRs [17,18,28,29,31,32]. Additionally, through *IGV* sequencing, N-gly motifs have been identified in over 95% of FL subclones at different disease stages [17,19]. These motifs can vary in specific amino acid sequences (N-X-S/T) between patients and even within related subclones in a single patient due to changes introduced by SHM (Figure 1) [17,18,28,29,31,32]. Notably, the N-gly motifs, which are rarely seen in healthy B cells, enable the attachment of oligomannose residues in FL cells, with no evidence of further modifications permitting the expression of mannosylated IgV on the FL cell surface [15,16,28,31]. The presence of mannosylated IgV in the BCR on the FL cell surface allows BCR activation through mannose/lectin interactions [13,15,16].

N-gly motifs have been shown to occur early in the development of disease and remain stable throughout disease progression, despite therapy [17,18]. The prevalence of N-gly motifs in FL cells varies among FL patients. Whereas most patients acquire at least one N-gly motif early in the disease, only a small proportion of patients do not exhibit this in either the heavy or light chains of the Ig (N-gly-negative) at any tumor site and time point [17]. Interestingly, there are cases with discordant N-gly motif patterns across different tumor sites or that show alterations in the sequence of motifs during disease progression, as has been observed by Odabashian (Figure 1) [17,18]. The persistence of N-gly motifs regardless of ongoing SHM indicates that the motifs confer a selective advantage to the cells and once introduced, ensure the survival and proliferation of the FL cells. The random nature of SHM allows for point mutations to alter the sequence, removing the N-gly motifs from *IgV*. However, the scarcity of N-gly-negative subclones, with frequencies such as 1.6% to 2.7% in the subclone population across various disease events for specific patients and their subsequent deletion evident from the analysis of longitudinal samples, suggests that N-gly-positive subclones expand preferentially. This scarcity underlies the functional importance of IgV N-glycosylation in FL pathogenesis, highlighting it as potential therapeutic targets in FL pathogenesis, highlighting them as potential therapeutic targets [13,17,18].

Mutations introduced by SHM that affect the N-gly motifs or nearby amino acids can theoretically result in four scenarios. First, the N-gly motif remains and contains oligomannose. Second, the N-gly motif remains but is unable to be glycosylated. Third, the N-gly motif remains but is inaccessible to oligomannose-binding lectins due to steric changes caused by the composition of amino acids in the variable region. Fourth, the N-gly motif is lost. All these scenarios are likely reflected in transcriptional changes with pathobiological consequences, such as dynamic changes in the rate of SHM, subclone proliferation, retention, and a gradual or sudden loss of viability.

## 4. Outcomes of N-Glycosylation Loss in Follicular Lymphoma Immunoglobulin

The longitudinal analysis showed that the outright loss of the motif leads to the deletion of the N-gly-negative subclone in most of the cases. However, rare N-gly-negative subclones can temporarily maintain division and SHM (Figure 2) [17,18]. As shown in the figure, in the example of longitudinal samples from three patients, the N-gly-negative subclones can be tracked and some show evident proliferation and ongoing SHM. Notably, in several instances, the ongoing SHM was able to re-introduce an N-gly motif (Figure 2). The proportion of N-gly-negative subclones among the total subclonal range in a sample varied between 0.69% and 2.7% across patients and disease events in a study by Odabashian et al. For example, Patient 1 at diagnosis had 1.7% N-gly-negative subclones with 1.8% at transformation and Patient 6 had 1.6%, 2.1% and 2.7% N-gly-negative subclones across FL diagnosis, tFL diagnosis and tFL relapse, respectively.

In a single-cell longitudinal analysis of 17 patients, Haebe et al. observed that 77% of patients in their cohort had acquired N-gly motifs in IgV but that two patients presented >97% of N-gly-negative clones across tissue sites and timepoints. There were also noticeable site differences in two other patients, with some tissue sites containing mostly negative subclones but other sites containing mostly positive subclones [14]. Although N-gly-positive subclones were prevalent in most patients, N-gly-negative clones demonstrated the ability to survive, at least within the time points covered in this study.

FL Ig N-gly influences TME interactions, and its loss can reduce the survival signals that FL cells receive from oligomannose-binding proteins through antigen-independent BCR activation. This disruption can affect the overall growth and persistence of the FL subclones. Although the majority of FL subclones retain N-gly motifs early and throughout the disease, the identification of N-gly-negative subclones represents an insufficiently explored aspect of FL [14,17,18,23]. These rare subclones present a unique opportunity to identify the supportive signal that sustains their proliferation without relying purely on N-gly-dependent BCR activation. Our current understanding of FL clonal evolution places emphasis on the importance of N-gly-mediated BCR signaling for survival and proliferation. However, the existence of these transiently proliferating N-gly-negative subclones indicates that there are alternative pathways that can also support FL progression (Figure 3).

Recent attempts to evaluate the consequences of N-gly loss have highlighted the involvement of alternative metabolic pathways and a shift towards a light zone (LZ) phenotype in negative subclones. N-gly-negative FL subclones have been observed to upregulate certain metabolic pathways, such as glycolysis, oxidative phosphorylation, fatty acid metabolism and MTORC1 signaling, which enable them to survive and proliferate transiently, despite presumably absent BCR signaling [14]. In their single-cell analysis of 17 longitudinal samples, Haebe et al. showed that N-gly-negative subclones can persist and even expand under specific circumstances, particularly when metabolic pathways such as oxidative phosphorylation and glycolysis are upregulated due to an increase in the activity of metabolism-regulating genes (*LDHA, COX5A, SLC1A4*). In contrast, N-gly-positive subclones predictably showed an increased reliance on BCR-associated genes such as *CD79B, BANK* and *BTK*. This suggests that the proliferating N-gly-negative subclones may employ a competitive survival mechanism distinct from N-gly-positive subclones [14]. The N-gly-negative subclones might also rely on other alternative pathways for survival. In a scRNA analysis of 1214 single cells obtained from an untreated FL case, it was shown that the gene expression profiles of N-gly-negative FL cells resemble LZ germinal center (GC) B cells, including the upregulation of CD83, a marker of transition from the dark zone (DZ). Meanwhile, N-gly-positive cells exhibited features of DZ GC cells, with the upregulation of *FOXO1* and *AICDA,* implying active SHM [19].

## 5. N-Gly-Positive Non-Proliferating Subclones

Interestingly, the analysis of FL subclones reveals that certain N-gly-positive subclones evidently do not demonstrate further SHM (Figure 1) [17]. This intriguing observation may have several explanations. In the context of N-glycosylation, it raises the question of whether the lectin interaction involving the FL IgV/BCR in these subclones is compromised due to possible steric alterations in the IgV regions, with these induced by SHM changes affecting the amino acid sequence. Such steric alterations may also have effects on the N-gly motif’s neighboring amino acid properties, potentially either hindering glycosylation or impeding the access of mannose-binding proteins to the oligomannose on the IgV chains [32,33].

Additionally, the observation of non-proliferating N-gly-positive subclones raises the question of whether the survival, proliferation and SHM of N-gly FL subclones are underlined by distinct mechanisms. This suggests unique survival strategies, differential interactions within the TME, and distinct responses to the treatment of non-proliferating N-gly-positive subclones compared to their proliferating counterparts. A further investigation of FL IgV steric modifications, coupled with a comparative analysis of transcriptional signatures and the mutational load between proliferating N-gly-positive and N-gly-negative subclones, could shed the light on the above questions.

## 6. Other B-Cell Lymphomas

Acquired N-gly is the hallmark of FL and is rarely seen in healthy B cells or in other common B-cell malignancies such as chronic lymphocytic leukemia or myeloma. However, in several GC-derived lymphomas, the incidence of acquired N-gly is high, raising questions about its role in pathogenesis and clonal evolution, and therefore the possibility of therapeutic targeting. Indeed, an analysis of other GC-derived lymphomas revealed heterogeneity in the presence of N-gly motifs in their IgVs. For example, in mucosa-associated lymphoid tissue (MALT) lymphoma, N-gly motifs are rarely found; however, a high prevalence of acquired IgV N-glycosylation has been observed in subtypes of DLBCL and Burkitt’s Lymphoma (BL). Specifically, *IgV* sequencing analysis in DLBCL identified that 60% of GCB and 13% of ABC-DLBCL carry N-gly motifs with an increased presence in FRs, especially in ABC-DLBCL [34]. In BL, 82% of endemic and 43% of sporadic BL cases had N-gly motifs, with 50% found in CDR [13,35]. The distinct localization of N-gly motifs in FL, with CDR selection at ~80%, contrasts with the prevalence of N-gly motifs in FRs detected in subtypes of DLBCL and BL. The selection of the N-gly motifs suggests that they have potential roles in the pathogenesis of DLBCL and BL, but their differential distribution likely reflects disease-specific mechanisms. Further research is needed to elucidate these mechanisms, but the approaches proposed in the “Future Directions” section are likely to be useful in studying lymphomas where IgV N-gly have been found.

## 7. Conclusions

FL demonstrates extensive clonal evolution driven by TME interactions in combination with genetic and epigenetic alterations. To date, longitudinal studies of clonal evolution in FL have shed light on the role of the conserved N-gly motifs in FL subclones, which are required for progression and proliferation, at least until transformation into more aggressive DLBCL [34]. In FL, these motifs occur early in disease development and are positively selected during FL evolution. They are evidently essential for pathogenesis, enabling FL BCR activation through mannose/lectin interactions.

The loss of N-gly motifs leads to the subclonal deletion of negative FL cells. However, recent studies have identified N-gly-negative subclones that are capable of proliferating, though usually these are short lived. The presence and proliferative capabilities of N-gly-negative subclones in FL provides an important avenue for the investigation of BCR-independent survival mechanisms. Recent studies have suggested that N-gly-negative subclones may rely on alternative pro-survival metabolic pathways or adopt distinct transcription profiles, independent of traditional BCR signaling (Figure 3) [14,17,19]. Additionally, the identification of N-gly-positive subclones that are also short lived implies the existence of different survival strategies. The dichotomy between proliferating and non-proliferating subclones, either N-gly-positive or negative, indicates distinct transcriptional and functional dynamics that require further study.

## 8. Future Directions

A broader approach to clonal evolution studies of FL is essential, particularly one that follows the N-gly status of the FL subclones and their capacity for proliferation and SHM. This comprehensive analysis aims to gain a more critical understanding of FL pathobiology and provide a framework for identifying novel therapeutic targets for improved disease management.

Firstly, a focused investigation comparing N-gly-positive proliferating major clones with N-gly-positive short-lived subclones as well as N-gly-negative non-proliferating and proliferating subclones is essential. Recent publications by Odabashian, Haebe and van Bergen have outlined promising directions. By determining the pathways that govern proliferation, SHM and survival across subclones with distinct IgV N-gly statuses in integrated longitudinal analyses of a multicentered cohort of FL patients, we can uncover critical insights into FL pathobiology. This will be required for the proposed therapeutic targeting of the N-gly/lectin interaction to succeed. The loss of the N-gly motif may reduce the effectiveness of such treatments or lead to the selection of certain N-gly-negative subclones. Additional targeting of the alternative metabolic pathways may prove effective [13,36].

Secondly, the analysis of the SHM-dependent steric alterations in the amino acid sequences surrounding N-gly motifs warrants further study. These changes may potentially alter the magnitude of lectin-dependent BCR activation and its duration or render it inactive. Recent advances in the development of new *in silico* analytical methods for modeling protein structure and interaction could reveal how variations in amino acid composition affect glycosylation and lectin binding. A systematic re-evaluation of existing and new datasets could uncover alternative mechanisms driving aggressive or treatment-resistant subclones, with an expectation of developing innovative therapeutic strategies that disrupt these alternative mechanisms to improve FL management.

Finally, similar investigations in other GC-derived lymphomas with the known selection of N-gly motifs could explain the explain the role of IgV N-glycosylation in diseases such as DLBCL and BL.

## Figures and Tables

**Figure 1 cancers-17-01219-f001:**
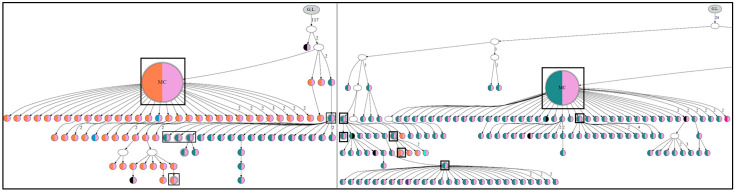
Lineage trees for a patient show the N-gly status of each subclone. The left panel represents the lineage tree of the 1st relapse event, whereas the right panel represents part of the lineage tree of the 2nd relapse event. Each subclone is represented by a node, and major clones are indicated in larger nodes. Nodes are split into two colors to represent the two N-gly motifs found in the FR2 and CDR3 regions. The FR2 site is represented in the left half and the CDR3 site is represented in the right half of the nodes. Nodes differing in color to the major clone represent subclones with different N-gly motif codon sequences. Black represents an absence of the FR2 or CDR3 site. Major clones between the two disease events have a different codon sequence in the FR2 N-gly site, indicated by the difference in color. Nodes in boxes represent the subclones shared between events. White nodes represent subclones inferred to exist but not detected through 454 sequencing. Germline nodes are in grey at the top of the tree, marked G.L. Numbers on branches indicate >1 mutation separating one node from the other. MC; major clone. Adapted from Odabashian et al. [17].

**Figure 2 cancers-17-01219-f002:**
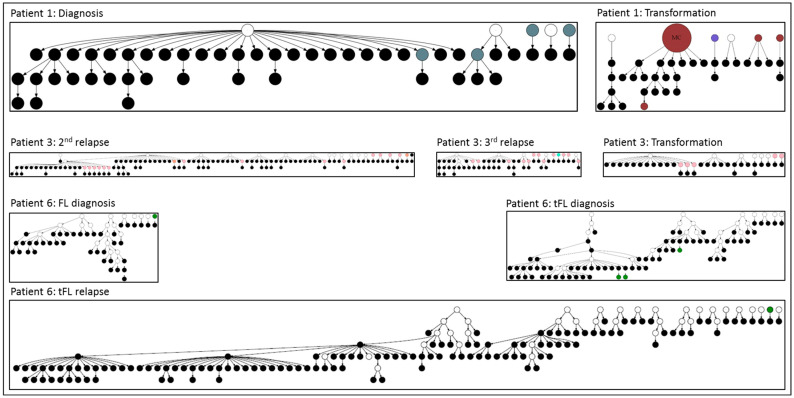
Relationship between N-gly-negative clones and their direct ancestral clone in FL patients. The negative subclones, parent clone and progenitor clone are shown in the figure. Black nodes represent N-gly-negative clones, white nodes represent undetected clones, and other node colors in the different patients represent N-gly-positive clones. Adapted from Odabashian et al. [17].

**Figure 3 cancers-17-01219-f003:**
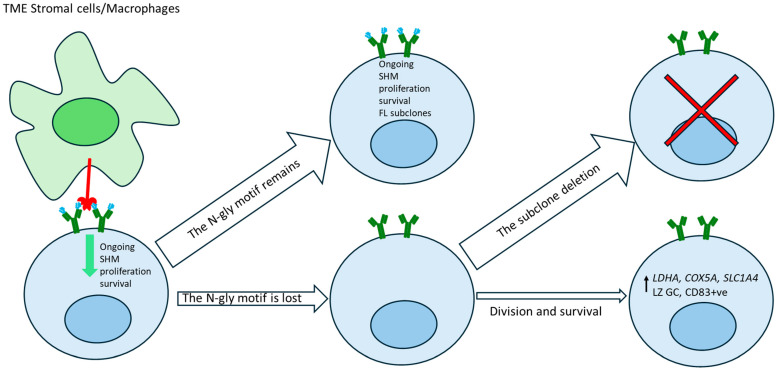
Impact of IgV N-glycosylation maintenance or loss on follicular lymphoma clonal evolution. N-gly oligomannose (blue) in the variable region of FL Immunoglobulin facilitates antigen-independent BCR activation through oligomannose–lectin interactions. DC-SIGN (red), expressed by stromal cells and macrophages in the tumor microenvironment (TME), promotes FL BCR activation, supporting FL cell survival, proliferation, and the persistence of N-gly-positive subclones. Somatic hypermutation (SHM) can lead to N-gly motif loss, resulting in the deletion of most N-gly-negative subclones. However, rare N-gly-negative subclones survive, likely due to upregulated oxidative phosphorylation and glycolysis. These metabolic adaptations may contribute to FL progression and the heterogeneity of subclones.

## Data Availability

The original data generated by M.O. and S.K. that has been presented in the study are openly available in the European Nucleotide Archive, BioProject PRJNA608656.

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
