# Peer review of "Perspective on Immunoglobulin N-Glycosylation Status in Follicular Lymphoma: Uncovering BCR-Dependent and Independent Mechanisms Driving Subclonal Evolution"

_cancers, 2025, doi:10.3390/cancers17071219_

Round 1
Reviewer 1 Report
Comments and Suggestions for Authors
Manu et al provide a detailed discussion on the role of N-glycosylation in follicular lymphoma that will be of interest and relevance to the field. Overall this is well written and informative. I have a few minor comments listed below which I believe should be addressed prior to publication.
- Provide a figure/schematic overview showing the interactions between FL cells, BCR, N-glyc and mannose/lectins
- The simple summary should be re-written to a more simple overview
- First paragraph the % survival rate at 10 years should be added
- The "certain genetic and epigenetic drives" should be explained in the second paragraph
- Initial drug response rate is mentioned however the standard of care should be explained/outlined here.
- Does N-glycosylation affect outcome to drug treatment in FL? This should be discussed
- A few typos were noticed throughout
Author Response
Dear Reviewers,
We sincerely thank you for your positive and constructive feedback on our manuscript. We are grateful for your suggestions, which have helped us to improve the quality of our manuscript.
Two of the four reviewers noted that the English language could be improved too and that some typos also should be addressed. Please note that the manuscript is written in UK English. In response, we have thoroughly edited the text to enhance readability and have corrected several typos. All revisions and additional text have been highlighted in yellow in the latest version of the manuscript for your convenience.
Please find our detailed responses to each of the reviewers’ comments below.
Reviewer 1
Comment 1:
- Provide a figure/schematic overview showing the interactions between FL cells, BCR, N-glyc and mannose/lectins
Response 1:
Added Figure 3
Comment 2:
- The simple summary should be re-written to a more simple overview
Response 2:
The lay summary has been further simplified
Comment 3:
- First paragraph the % survival rate at 10 years should be added
Response 3:
~80%, Added line 44; added additional ref 3
Comment 4:
- The "certain genetic and epigenetic drives" should be explained in the second paragraph
Response 4:
Added lines 54-55 and more discussion in lines 88 to 92, added additional ref 9
Comment 5:
- Initial drug response rate is mentioned however the standard of care should be explained/outlined here.
Response 5:
Added lines 46 to 50 and ref 4
Comment 6:
- Does N-glycosylation affect outcome to drug treatment in FL? This should be discussed
Response 6:
Virtually all FL cases (apart from rare ones) have N-gly in IgV. N-glycosylation plays a role in the persistence of FL clones by providing survival advantages even in the presence of treatment. This is the main evident effect of IgV glycosylation in FL so far. This has been discussed in the text lines 126-129. Importantly, blocking the mannose/lectin interaction with monoclonal antibodies or other means is under investigation as a potential therapeutic strategy. Lines 136-138 and 261-264.
Comment 7:
- A few typos were noticed throughout
Response 7:
Additional checks have been conducted, the manuscript is written in UK English
Reviewer 2
Comments and Suggestions for Authors
This is a timely review that describes how IgV N-gly motifs are acquired by SHM and its role in BCR activation and the pathogenesis of follicular lymphoma and would be of considerable interest in readers. Most N-gly negative clones are eliminated, but a small percentage can survive and proliferate. Negative clones have been shown to have a gene expression profile similar to light zone cells. The authors recommend pursuing transcriptomic profiling of these different clones to understand the role of N-gly in the context of pathogenic pathways in follicular lymphoma.
Comment 1:
Review is well written and the different sections are sequenced well. However the section describing the aim of the review is confusing. According to the authors, this perspective examines current literature and explores whether a detailed transcriptomic and functional comparison of FL subclones characterised by different N-gly status, proliferation and SHM rates will lead to comprehensive understanding of both N- gly-dependent and independent pro-survival and proliferative transcriptional signatures.
Since the data detailing the differences between FL subclones with different N-gly status and their effects on proliferation is not extensively explored in this review, the section should be modified to describe the actual content of the review. Role of future studies can be mentioned in the appropriate sections. The findings from Haebe et al regarding the gene expression profile can be described in more detail.
Response 1:
‘examines current literature’ part of the abstract has been modified ‘with a particular focus on N-gly-negative subclones’ to avoid confusion – line 34.
More discussion of Haebe results has been added. Lines 182-183, 186-188,
We would like to point out that since our manuscript (Odabashian, Blood, 2020) where we described the first observation of the rare N-gly-negative proliferating FL subclones there were only two publications that picked this discovery up and expanded our knowledge on N-gly-negative subclones’ behaviour (Haebe Blood 2023 and van Bergen Blood Adv 2023). The available literature is limited to these publications so far. This is the point we are making in this perspective – the FL field will benefit with more analysis of bigger multicentred FL cohorts based on the N-gly status and with particular focus on N-gly -negative proliferating subclones as these can instruct us on the major differences between BCR-dependent vs BCR-independent supportive pathways.
Comment 2:
Some of the references did not match the assertions mentioned. For eg –
the statement ‘N-gly negative FL subclones have been observed to upregulate certain metabolic pathways, to enable them to survive and proliferate transiently, despite presumably absent BCR signalling. [11].
Another instance:
In their single cell analysis of 17 longitudinal samples, Haebe et al have shown that N-gly negative subclones can persist and even expand under specific circumstances, particularly when met- abolic pathways such as oxidative phosphorylation and glycolysis are upregulated due to an increase activity of LDHA, COX5A, SLC1A4 genes. This suggests they may have a competitive survival mechanism distinct from N-gly positive subclones [11].
Response 2:
In these two instances, we discussed results from Haebe’s manuscript, which was ref 11 and is now ref 14 in the latest version. In their manuscript, Haebe et al. analysed 17 longitudinal FL samples and compared expression profiles of N-gly positive vs negative FL subclones. They observed the upregulation of metabolic pathways reflected in the upregulation of LDHA, COX5A, and SLC1A4 in negative clones and BCR-related pathways in positive (BTK, CD79B).
Comment 3:
Please ensure all references quoted are accurate.
Response 3:
All were now double checked
Comments on the Quality of English Language
OK
Reviewer 3
Comments and Suggestions for Authors
Comment 1:
- FL is characterized by acquired N-glycosylation, including oligomannose, in the immunoglobulin variable region. Please list its functions. Please explain why it is not present in other indolent B cell tumors. Please also explain why it is present in BL and DLBCL.
Response 1:
We and others have shown the role on the unusual N-gly of FL IgVs in the antigen-independent activation of FL BCR through lectin/mannose interaction that is enough for BCR crosslinking and activation of downstream signalling. Similar motifs can be found in some other B-cell malignancies which has been discussed, please see lines 222-236 In short, the B cell malignancies with a likely antigen dependent activation of BCR (MALT, CLL) very rarely have N-gly motifs in IgV, the same as in healthy B cells. The GC derived lymphomas have higher N-gly prevalence with FL the highest. The differences in the N-gly location (CDR vs FR) in FL vs DLBCL and BL warrant further investigation and currently has no satisfactory explanation (distinct origin, selection pressures, mutation load?).
Comment 2:
- Are there cases without acquired N-glycosylation in the immunoglobulin variable region where there is no N-acetylsylation? Please explain.
Response 2:
As mentioned in the manuscript there are rare clinical cases without N-gly, e.g. in Haebe Blood 2023, one patient had negative FL clones whereas the other three had high prevalence of negative subclones. To reiterate, these are exceptional cases. These unique cases as well as the cases where N-gly-negative subclones are rare but still can be found, provide the opportunity to explore and contrast the BCR-dependent vs BCR-independent survival strategies. The aim of this manuscript is to encourage the FL community to exploit this opportunity.
Comment 3:
- Since N-glyconegative subclones persist temporarily, have alternative survival mechanisms been elucidated? Please explain.
Response 3:
In this manuscript we explain that the FL community is at the early stage of exploring your question. It has been 4 years since these subclones were observed for the first time. Haebe Blood 2023 and van Bergen Blood Adv 2023 are first two follow up studies that highlighted changes in the survival strategy in N-gly-negative subclones with activation of glycolysis, oxidative phosphorylation, fatty acid metabolism, MTORC1 signalling as well as assuming more of LZ B-cell expression profile. This perspective discusses the alternative pathways uncovered by Haebe and van Bergen. We are anxious to see more of these studies and urge the experts in the field to investigate this problem further.
Comment 4:
- Although the initial drug response rate of FL is high, there is a problem with resistance to existing drugs, and transformation to DLBCL is cited as one of the reasons why more than 28% of patients are resistant to treatment. Is N-glycosylation, including oligomannose, in the immunoglobulin variable region enhanced before and after transformation to DLBCL? Please explain.
Response 4:
The current multicentred results imply that N-glycosylation IgV region remains stable before and after transformation from FL to tFL or DLBCL. But more longitudinal paired samples analysis still required for the definite answer to your question.
Reviewer 4
Comments and Suggestions for Authors
In this article, the authors propose to uncover some of the mechanisms underlying the clonal evolution of follicular B-cell lymphoma. This article is interesting because it highlights protein-carbohydrate interactions and the importance of N-glycosylation alterations in the pathophysiology of follicular lymphoma, as is likely the case in many other cancers. These protein-oligosaccharide interactions involve various lectin-like molecules, such as DC-SIGN, a molecule discussed in this article. These lectin-oligosaccharide interactions are particularly involved in antitumor immune defense and, more generally, in the tumor microenvironment.
The mechanisms discussed concern mutations allowing "the introduction of N-glycosylation (N-gly) amino acid sequence motifs into the immunoglobulin (IgV) variable region by continuous somatic hypermutation (SHM)". These N-gly motifs, carrying oligomannose motifs capable of interacting with lectin-type molecules, may be involved in the survival of malignant clones. The authors cite numerous clinical observations indicating that the presence of hyper-mannosylated motifs at the BCR level could be involved in the survival of N-gly-positive clones. The situation is more complex at the level of N-gly-negative clones, which would be eliminated except for certain minority clones that would survive by involving mechanisms to be discovered. This suggests that while BCR hyper-mannosylation likely plays an important role in the growth of follicular lymphoma, there are other mechanisms involved in N-gly-negative clones. Indeed, independently with possible new mechanisms specifically involved in N-gly-negative clones survivale, other mechanisms involving various mutations and epigenetic modifications which well documented in the literature, could be involved in the N-gly-negative clones survivale.
On the other hand, can the authors answer these few questions?
Comment 1:
1- What is the structure(s) of the hyper-mannosylated oligosaccharides found at the N-gly sequences of the BCR?
Response 1:
Added text in lines 58-59
Comment 2:
2- Is there heterogeneity in the oligosaccharide sequences carried by the N-gly motifs of the BCR of clones within a tumor site?
Response 2:
Added discussion in line 60
Comment 3:
3- Is there variability in the oligosaccharide motifs of the BCR between different patients?
Response 3:
We assume the question is about N-gly sequence motif. There is. The N-gly motif sequence N-X-S/T can be lost (e.g. S/T is lost) or changed (e.g. different N-X introduced by SHM – the change of the motif but N-gly remains) by ongoing SHM. The N-gly motif sequence N-X-S/T can vary between patients and within FL in some cases. This fact has been discussed. Added text in lines 120-122 in the text and Figure 1 (an example of the variability withing a single patient, Odabashian, Blood, 2020).
Comment 4:
4- Is there hyper-mannosylation at the level of the N-glycans of other glycoproteins of malignant B lymphocytes?
Response 4:
It is a very interesting question and is possible but outside of this review scope. The main point of this review is the role of the antigen-independent BCR activation through mannose/lectin interaction and how the outcome for FL subclones changes when glycosylation motif and therefore unusual FL IgV N-gly is lost.
Comment 5:
Finally, to facilitate understanding and strengthen your perspective, it might be helpful to provide readers with:
- a table outlining other mutations and epigenetic modifications that may play a role in oncogenesis and pathogenicity (e.g., KMT2D, RRAGC, TNFAIP3, TNFRSF14, FOXO1, etc.). In this regard, the results obtained in the EPIC (500,000 individuals) and CPS (120,000 individuals) cohorts show that the association of the t(14-18) translocation with the epigenetic modification CREBBP could be the precursor necessary for cell cancerization (J. Schroers-Martin et al. Tracing Founder Mutations in Circulating and Tissue-Resident Follicular Lymphoma Precursors. Cancer Discov, 2023; doi: 10.1158/2159–8290.CD-23–0111)
Response 5:
Added in the text, reference 9, added lines 88 – 92
Comment 6:
- b)a simple diagram reminiscent of the lymphoma tumor microenvironment content, with principal cell and molecule interactions (BCR/DC-SIGN…).
Response 6:
Added Figure 3
Reviewer 2 Report
Comments and Suggestions for Authors
This is a timely review that describes how IgV N-gly motifs are acquired by SHM and its role in BCR activation and the pathogenesis of follicular lymphoma and would be of considerable interest in readers. Most N-gly negative clones are eliminated, but a small percentage can survive and proliferate. Negative clones have been shown to have a gene expression profile similar to light zone cells. The authors recommend pursuing transcriptomic profiling of these different clones to understand the role of N-gly in the context of pathogenic pathways in follicular lymphoma.
Review is well written and the different sections are sequenced well. However the section describing the aim of the review is confusing. According to the authors, this perspective examines current literature and explores whether a detailed transcriptomic and functional comparison of FL subclones characterised by different N-gly status, proliferation and SHM rates will lead to comprehensive understanding of both N- gly-dependent and independent pro-survival and proliferative transcriptional signatures.
Since the data detailing the differences between FL subclones with different N-gly status and their effects on proliferation is not extensively explored in this review, the section should be modified to describe the actual content of the review. Role of future studies can be mentioned in the appropriate sections. The findings from Haebe et al regarding the gene expression profile can be described in more detail.
Some of the references did not match the assertions mentioned. For eg –
the statement ‘N-gly negative FL subclones have been observed to upregulate certain metabolic pathways, to enable them to survive and proliferate transiently, despite presumably absent BCR signalling. [11].
Another instance:
In their single cell analysis of 17 longitudinal samples, Haebe et al have shown that N-gly negative subclones can persist and even expand under specific circumstances, particularly when met- abolic pathways such as oxidative phosphorylation and glycolysis are upregulated due to an increase activity of LDHA, COX5A, SLC1A4 genes. This suggests they may have a competitive survival mechanism distinct from N-gly positive subclones [11].
Please ensure all references quoted are accurate.
Comments on the Quality of English Language
OK
Author Response
Dear Reviewers,
We sincerely thank you for your positive and constructive feedback on our manuscript. We are grateful for your suggestions, which have helped us to improve the quality of our manuscript.
Two of the four reviewers noted that the English language could be improved too and that some typos also should be addressed. Please note that the manuscript is written in UK English. In response, we have thoroughly edited the text to enhance readability and have corrected several typos. All revisions and additional text have been highlighted in yellow in the latest version of the manuscript for your convenience.
Please find our detailed responses to each of the reviewers’ comments below.
Reviewer 2
Comments and Suggestions for Authors
This is a timely review that describes how IgV N-gly motifs are acquired by SHM and its role in BCR activation and the pathogenesis of follicular lymphoma and would be of considerable interest in readers. Most N-gly negative clones are eliminated, but a small percentage can survive and proliferate. Negative clones have been shown to have a gene expression profile similar to light zone cells. The authors recommend pursuing transcriptomic profiling of these different clones to understand the role of N-gly in the context of pathogenic pathways in follicular lymphoma.
Comment 1:
Review is well written and the different sections are sequenced well. However the section describing the aim of the review is confusing. According to the authors, this perspective examines current literature and explores whether a detailed transcriptomic and functional comparison of FL subclones characterised by different N-gly status, proliferation and SHM rates will lead to comprehensive understanding of both N- gly-dependent and independent pro-survival and proliferative transcriptional signatures.
Since the data detailing the differences between FL subclones with different N-gly status and their effects on proliferation is not extensively explored in this review, the section should be modified to describe the actual content of the review. Role of future studies can be mentioned in the appropriate sections. The findings from Haebe et al regarding the gene expression profile can be described in more detail.
Response 1:
‘examines current literature’ part of the abstract has been modified ‘with a particular focus on N-gly-negative subclones’ to avoid confusion – line 34.
More discussion of Haebe results has been added. Lines 182-183, 186-188,
We would like to point out that since our manuscript (Odabashian, Blood, 2020) where we described the first observation of the rare N-gly-negative proliferating FL subclones there were only two publications that picked this discovery up and expanded our knowledge on N-gly-negative subclones’ behaviour (Haebe Blood 2023 and van Bergen Blood Adv 2023). The available literature is limited to these publications so far. This is the point we are making in this perspective – the FL field will benefit with more analysis of bigger multicentred FL cohorts based on the N-gly status and with particular focus on N-gly -negative proliferating subclones as these can instruct us on the major differences between BCR-dependent vs BCR-independent supportive pathways.
Comment 2:
Some of the references did not match the assertions mentioned. For eg –
the statement ‘N-gly negative FL subclones have been observed to upregulate certain metabolic pathways, to enable them to survive and proliferate transiently, despite presumably absent BCR signalling. [11].
Another instance:
In their single cell analysis of 17 longitudinal samples, Haebe et al have shown that N-gly negative subclones can persist and even expand under specific circumstances, particularly when met- abolic pathways such as oxidative phosphorylation and glycolysis are upregulated due to an increase activity of LDHA, COX5A, SLC1A4 genes. This suggests they may have a competitive survival mechanism distinct from N-gly positive subclones [11].
Response 2:
In these two instances, we discussed results from Haebe’s manuscript, which was ref 11 and is now ref 14 in the latest version. In their manuscript, Haebe et al. analysed 17 longitudinal FL samples and compared expression profiles of N-gly positive vs negative FL subclones. They observed the upregulation of metabolic pathways reflected in the upregulation of LDHA, COX5A, and SLC1A4 in negative clones and BCR-related pathways in positive (BTK, CD79B).
Comment 3:
Please ensure all references quoted are accurate.
Response 3:
All were now double checked
Comments on the Quality of English Language
OK
Reviewer 3
Comments and Suggestions for Authors
Comment 1:
- FL is characterized by acquired N-glycosylation, including oligomannose, in the immunoglobulin variable region. Please list its functions. Please explain why it is not present in other indolent B cell tumors. Please also explain why it is present in BL and DLBCL.
Response 1:
We and others have shown the role on the unusual N-gly of FL IgVs in the antigen-independent activation of FL BCR through lectin/mannose interaction that is enough for BCR crosslinking and activation of downstream signalling. Similar motifs can be found in some other B-cell malignancies which has been discussed, please see lines 222-236 In short, the B cell malignancies with a likely antigen dependent activation of BCR (MALT, CLL) very rarely have N-gly motifs in IgV, the same as in healthy B cells. The GC derived lymphomas have higher N-gly prevalence with FL the highest. The differences in the N-gly location (CDR vs FR) in FL vs DLBCL and BL warrant further investigation and currently has no satisfactory explanation (distinct origin, selection pressures, mutation load?).
Comment 2:
- Are there cases without acquired N-glycosylation in the immunoglobulin variable region where there is no N-acetylsylation? Please explain.
Response 2:
As mentioned in the manuscript there are rare clinical cases without N-gly, e.g. in Haebe Blood 2023, one patient had negative FL clones whereas the other three had high prevalence of negative subclones. To reiterate, these are exceptional cases. These unique cases as well as the cases where N-gly-negative subclones are rare but still can be found, provide the opportunity to explore and contrast the BCR-dependent vs BCR-independent survival strategies. The aim of this manuscript is to encourage the FL community to exploit this opportunity.
Comment 3:
- Since N-glyconegative subclones persist temporarily, have alternative survival mechanisms been elucidated? Please explain.
Response 3:
In this manuscript we explain that the FL community is at the early stage of exploring your question. It has been 4 years since these subclones were observed for the first time. Haebe Blood 2023 and van Bergen Blood Adv 2023 are first two follow up studies that highlighted changes in the survival strategy in N-gly-negative subclones with activation of glycolysis, oxidative phosphorylation, fatty acid metabolism, MTORC1 signalling as well as assuming more of LZ B-cell expression profile. This perspective discusses the alternative pathways uncovered by Haebe and van Bergen. We are anxious to see more of these studies and urge the experts in the field to investigate this problem further.
Comment 4:
- Although the initial drug response rate of FL is high, there is a problem with resistance to existing drugs, and transformation to DLBCL is cited as one of the reasons why more than 28% of patients are resistant to treatment. Is N-glycosylation, including oligomannose, in the immunoglobulin variable region enhanced before and after transformation to DLBCL? Please explain.
Response 4:
The current multicentred results imply that N-glycosylation IgV region remains stable before and after transformation from FL to tFL or DLBCL. But more longitudinal paired samples analysis still required for the definite answer to your question.
Reviewer 4
Comments and Suggestions for Authors
In this article, the authors propose to uncover some of the mechanisms underlying the clonal evolution of follicular B-cell lymphoma. This article is interesting because it highlights protein-carbohydrate interactions and the importance of N-glycosylation alterations in the pathophysiology of follicular lymphoma, as is likely the case in many other cancers. These protein-oligosaccharide interactions involve various lectin-like molecules, such as DC-SIGN, a molecule discussed in this article. These lectin-oligosaccharide interactions are particularly involved in antitumor immune defense and, more generally, in the tumor microenvironment.
The mechanisms discussed concern mutations allowing "the introduction of N-glycosylation (N-gly) amino acid sequence motifs into the immunoglobulin (IgV) variable region by continuous somatic hypermutation (SHM)". These N-gly motifs, carrying oligomannose motifs capable of interacting with lectin-type molecules, may be involved in the survival of malignant clones. The authors cite numerous clinical observations indicating that the presence of hyper-mannosylated motifs at the BCR level could be involved in the survival of N-gly-positive clones. The situation is more complex at the level of N-gly-negative clones, which would be eliminated except for certain minority clones that would survive by involving mechanisms to be discovered. This suggests that while BCR hyper-mannosylation likely plays an important role in the growth of follicular lymphoma, there are other mechanisms involved in N-gly-negative clones. Indeed, independently with possible new mechanisms specifically involved in N-gly-negative clones survivale, other mechanisms involving various mutations and epigenetic modifications which well documented in the literature, could be involved in the N-gly-negative clones survivale.
On the other hand, can the authors answer these few questions?
Comment 1:
1- What is the structure(s) of the hyper-mannosylated oligosaccharides found at the N-gly sequences of the BCR?
Response 1:
Added text in lines 58-59
Comment 2:
2- Is there heterogeneity in the oligosaccharide sequences carried by the N-gly motifs of the BCR of clones within a tumor site?
Response 2:
Added discussion in line 60
Comment 3:
3- Is there variability in the oligosaccharide motifs of the BCR between different patients?
Response 3:
We assume the question is about N-gly sequence motif. There is. The N-gly motif sequence N-X-S/T can be lost (e.g. S/T is lost) or changed (e.g. different N-X introduced by SHM – the change of the motif but N-gly remains) by ongoing SHM. The N-gly motif sequence N-X-S/T can vary between patients and within FL in some cases. This fact has been discussed. Added text in lines 120-122 in the text and Figure 1 (an example of the variability withing a single patient, Odabashian, Blood, 2020).
Comment 4:
4- Is there hyper-mannosylation at the level of the N-glycans of other glycoproteins of malignant B lymphocytes?
Response 4:
It is a very interesting question and is possible but outside of this review scope. The main point of this review is the role of the antigen-independent BCR activation through mannose/lectin interaction and how the outcome for FL subclones changes when glycosylation motif and therefore unusual FL IgV N-gly is lost.
Comment 5:
Finally, to facilitate understanding and strengthen your perspective, it might be helpful to provide readers with:
- a table outlining other mutations and epigenetic modifications that may play a role in oncogenesis and pathogenicity (e.g., KMT2D, RRAGC, TNFAIP3, TNFRSF14, FOXO1, etc.). In this regard, the results obtained in the EPIC (500,000 individuals) and CPS (120,000 individuals) cohorts show that the association of the t(14-18) translocation with the epigenetic modification CREBBP could be the precursor necessary for cell cancerization (J. Schroers-Martin et al. Tracing Founder Mutations in Circulating and Tissue-Resident Follicular Lymphoma Precursors. Cancer Discov, 2023; doi: 10.1158/2159–8290.CD-23–0111)
Response 5:
Added in the text, reference 9, added lines 88 – 92
Comment 6:
- b)a simple diagram reminiscent of the lymphoma tumor microenvironment content, with principal cell and molecule interactions (BCR/DC-SIGN…).
Response 6:
Added Figure 3
Reviewer 1
Comment 1:
- Provide a figure/schematic overview showing the interactions between FL cells, BCR, N-glyc and mannose/lectins
Response 1:
Added Figure 3
Comment 2:
- The simple summary should be re-written to a more simple overview
Response 2:
The lay summary has been further simplified
Comment 3:
- First paragraph the % survival rate at 10 years should be added
Response 3:
~80%, Added line 44; added additional ref 3
Comment 4:
- The "certain genetic and epigenetic drives" should be explained in the second paragraph
Response 4:
Added lines 54-55 and more discussion in lines 88 to 92, added additional ref 9
Comment 5:
- Initial drug response rate is mentioned however the standard of care should be explained/outlined here.
Response 5:
Added lines 46 to 50 and ref 4
Comment 6:
- Does N-glycosylation affect outcome to drug treatment in FL? This should be discussed
Response 6:
Virtually all FL cases (apart from rare ones) have N-gly in IgV. N-glycosylation plays a role in the persistence of FL clones by providing survival advantages even in the presence of treatment. This is the main evident effect of IgV glycosylation in FL so far. This has been discussed in the text lines 126-129. Importantly, blocking the mannose/lectin interaction with monoclonal antibodies or other means is under investigation as a potential therapeutic strategy. Lines 136-138 and 261-264.
Comment 7:
- A few typos were noticed throughout
Response 7:
Additional checks have been conducted, the manuscript is written in UK English
Reviewer 3 Report
Comments and Suggestions for Authors
- FL is characterized by acquired N-glycosylation, including oligomannose, in the immunoglobulin variable region. Please list its functions. Please explain why it is not present in other indolent B cell tumors. Please also explain why it is present in BL and DLBCL.
- Are there cases without acquired N-glycosylation in the immunoglobulin variable region where there is no N-acetylsylation? Please explain.
- Since N-glyconegative subclones persist temporarily, have alternative survival mechanisms been elucidated? Please explain.
- Although the initial drug response rate of FL is high, there is a problem with resistance to existing drugs, and transformation to DLBCL is cited as one of the reasons why more than 28% of patients are resistant to treatment. Is N-glycosylation, including oligomannose, in the immunoglobulin variable region enhanced before and after transformation to DLBCL? Please explain.
- FL is characterized by acquired N-glycosylation, including oligomannose, in the immunoglobulin variable region. Please list its functions. Please explain why it is not present in other indolent B cell tumors. Please also explain why it is present in BL and DLBCL.
- Are there cases without acquired N-glycosylation in the immunoglobulin variable region where there is no N-acetylsylation? Please explain.
- Since N-glyconegative subclones persist temporarily, have alternative survival mechanisms been elucidated? Please explain.
- Although the initial drug response rate of FL is high, there is a problem with resistance to existing drugs, and transformation to DLBCL is cited as one of the reasons why more than 28% of patients are resistant to treatment. Is N-glycosylation, including oligomannose, in the immunoglobulin variable region enhanced before and after transformation to DLBCL? Please explain.
Author Response
Dear Reviewers,
We sincerely thank you for your positive and constructive feedback on our manuscript. We are grateful for your suggestions, which have helped us to improve the quality of our manuscript.
Two of the four reviewers noted that the English language could be improved too and that some typos also should be addressed. Please note that the manuscript is written in UK English. In response, we have thoroughly edited the text to enhance readability and have corrected several typos. All revisions and additional text have been highlighted in yellow in the latest version of the manuscript for your convenience.
Please find our detailed responses to each of the reviewers’ comments below.
Reviewer 3
Comments and Suggestions for Authors
Comment 1:
- FL is characterized by acquired N-glycosylation, including oligomannose, in the immunoglobulin variable region. Please list its functions. Please explain why it is not present in other indolent B cell tumors. Please also explain why it is present in BL and DLBCL.
Response 1:
We and others have shown the role on the unusual N-gly of FL IgVs in the antigen-independent activation of FL BCR through lectin/mannose interaction that is enough for BCR crosslinking and activation of downstream signalling. Similar motifs can be found in some other B-cell malignancies which has been discussed, please see lines 222-236 In short, the B cell malignancies with a likely antigen dependent activation of BCR (MALT, CLL) very rarely have N-gly motifs in IgV, the same as in healthy B cells. The GC derived lymphomas have higher N-gly prevalence with FL the highest. The differences in the N-gly location (CDR vs FR) in FL vs DLBCL and BL warrant further investigation and currently has no satisfactory explanation (distinct origin, selection pressures, mutation load?).
Comment 2:
- Are there cases without acquired N-glycosylation in the immunoglobulin variable region where there is no N-acetylsylation? Please explain.
Response 2:
As mentioned in the manuscript there are rare clinical cases without N-gly, e.g. in Haebe Blood 2023, one patient had negative FL clones whereas the other three had high prevalence of negative subclones. To reiterate, these are exceptional cases. These unique cases as well as the cases where N-gly-negative subclones are rare but still can be found, provide the opportunity to explore and contrast the BCR-dependent vs BCR-independent survival strategies. The aim of this manuscript is to encourage the FL community to exploit this opportunity.
Comment 3:
- Since N-glyconegative subclones persist temporarily, have alternative survival mechanisms been elucidated? Please explain.
Response 3:
In this manuscript we explain that the FL community is at the early stage of exploring your question. It has been 4 years since these subclones were observed for the first time. Haebe Blood 2023 and van Bergen Blood Adv 2023 are first two follow up studies that highlighted changes in the survival strategy in N-gly-negative subclones with activation of glycolysis, oxidative phosphorylation, fatty acid metabolism, MTORC1 signalling as well as assuming more of LZ B-cell expression profile. This perspective discusses the alternative pathways uncovered by Haebe and van Bergen. We are anxious to see more of these studies and urge the experts in the field to investigate this problem further.
Comment 4:
- Although the initial drug response rate of FL is high, there is a problem with resistance to existing drugs, and transformation to DLBCL is cited as one of the reasons why more than 28% of patients are resistant to treatment. Is N-glycosylation, including oligomannose, in the immunoglobulin variable region enhanced before and after transformation to DLBCL? Please explain.
Response 4:
The current multicentred results imply that N-glycosylation IgV region remains stable before and after transformation from FL to tFL or DLBCL. But more longitudinal paired samples analysis still required for the definite answer to your question.
Reviewer 4
Comments and Suggestions for Authors
In this article, the authors propose to uncover some of the mechanisms underlying the clonal evolution of follicular B-cell lymphoma. This article is interesting because it highlights protein-carbohydrate interactions and the importance of N-glycosylation alterations in the pathophysiology of follicular lymphoma, as is likely the case in many other cancers. These protein-oligosaccharide interactions involve various lectin-like molecules, such as DC-SIGN, a molecule discussed in this article. These lectin-oligosaccharide interactions are particularly involved in antitumor immune defense and, more generally, in the tumor microenvironment.
The mechanisms discussed concern mutations allowing "the introduction of N-glycosylation (N-gly) amino acid sequence motifs into the immunoglobulin (IgV) variable region by continuous somatic hypermutation (SHM)". These N-gly motifs, carrying oligomannose motifs capable of interacting with lectin-type molecules, may be involved in the survival of malignant clones. The authors cite numerous clinical observations indicating that the presence of hyper-mannosylated motifs at the BCR level could be involved in the survival of N-gly-positive clones. The situation is more complex at the level of N-gly-negative clones, which would be eliminated except for certain minority clones that would survive by involving mechanisms to be discovered. This suggests that while BCR hyper-mannosylation likely plays an important role in the growth of follicular lymphoma, there are other mechanisms involved in N-gly-negative clones. Indeed, independently with possible new mechanisms specifically involved in N-gly-negative clones survivale, other mechanisms involving various mutations and epigenetic modifications which well documented in the literature, could be involved in the N-gly-negative clones survivale.
On the other hand, can the authors answer these few questions?
Comment 1:
1- What is the structure(s) of the hyper-mannosylated oligosaccharides found at the N-gly sequences of the BCR?
Response 1:
Added text in lines 58-59
Comment 2:
2- Is there heterogeneity in the oligosaccharide sequences carried by the N-gly motifs of the BCR of clones within a tumor site?
Response 2:
Added discussion in line 60
Comment 3:
3- Is there variability in the oligosaccharide motifs of the BCR between different patients?
Response 3:
We assume the question is about N-gly sequence motif. There is. The N-gly motif sequence N-X-S/T can be lost (e.g. S/T is lost) or changed (e.g. different N-X introduced by SHM – the change of the motif but N-gly remains) by ongoing SHM. The N-gly motif sequence N-X-S/T can vary between patients and within FL in some cases. This fact has been discussed. Added text in lines 120-122 in the text and Figure 1 (an example of the variability withing a single patient, Odabashian, Blood, 2020).
Comment 4:
4- Is there hyper-mannosylation at the level of the N-glycans of other glycoproteins of malignant B lymphocytes?
Response 4:
It is a very interesting question and is possible but outside of this review scope. The main point of this review is the role of the antigen-independent BCR activation through mannose/lectin interaction and how the outcome for FL subclones changes when glycosylation motif and therefore unusual FL IgV N-gly is lost.
Comment 5:
Finally, to facilitate understanding and strengthen your perspective, it might be helpful to provide readers with:
- a table outlining other mutations and epigenetic modifications that may play a role in oncogenesis and pathogenicity (e.g., KMT2D, RRAGC, TNFAIP3, TNFRSF14, FOXO1, etc.). In this regard, the results obtained in the EPIC (500,000 individuals) and CPS (120,000 individuals) cohorts show that the association of the t(14-18) translocation with the epigenetic modification CREBBP could be the precursor necessary for cell cancerization (J. Schroers-Martin et al. Tracing Founder Mutations in Circulating and Tissue-Resident Follicular Lymphoma Precursors. Cancer Discov, 2023; doi: 10.1158/2159–8290.CD-23–0111)
Response 5:
Added in the text, reference 9, added lines 88 – 92
Comment 6:
- b)a simple diagram reminiscent of the lymphoma tumor microenvironment content, with principal cell and molecule interactions (BCR/DC-SIGN…).
Response 6:
Added Figure 3
Reviewer 1
Comment 1:
- Provide a figure/schematic overview showing the interactions between FL cells, BCR, N-glyc and mannose/lectins
Response 1:
Added Figure 3
Comment 2:
- The simple summary should be re-written to a more simple overview
Response 2:
The lay summary has been further simplified
Comment 3:
- First paragraph the % survival rate at 10 years should be added
Response 3:
~80%, Added line 44; added additional ref 3
Comment 4:
- The "certain genetic and epigenetic drives" should be explained in the second paragraph
Response 4:
Added lines 54-55 and more discussion in lines 88 to 92, added additional ref 9
Comment 5:
- Initial drug response rate is mentioned however the standard of care should be explained/outlined here.
Response 5:
Added lines 46 to 50 and ref 4
Comment 6:
- Does N-glycosylation affect outcome to drug treatment in FL? This should be discussed
Response 6:
Virtually all FL cases (apart from rare ones) have N-gly in IgV. N-glycosylation plays a role in the persistence of FL clones by providing survival advantages even in the presence of treatment. This is the main evident effect of IgV glycosylation in FL so far. This has been discussed in the text lines 126-129. Importantly, blocking the mannose/lectin interaction with monoclonal antibodies or other means is under investigation as a potential therapeutic strategy. Lines 136-138 and 261-264.
Comment 7:
- A few typos were noticed throughout
Response 7:
Additional checks have been conducted, the manuscript is written in UK English
Reviewer 2
Comments and Suggestions for Authors
This is a timely review that describes how IgV N-gly motifs are acquired by SHM and its role in BCR activation and the pathogenesis of follicular lymphoma and would be of considerable interest in readers. Most N-gly negative clones are eliminated, but a small percentage can survive and proliferate. Negative clones have been shown to have a gene expression profile similar to light zone cells. The authors recommend pursuing transcriptomic profiling of these different clones to understand the role of N-gly in the context of pathogenic pathways in follicular lymphoma.
Comment 1:
Review is well written and the different sections are sequenced well. However the section describing the aim of the review is confusing. According to the authors, this perspective examines current literature and explores whether a detailed transcriptomic and functional comparison of FL subclones characterised by different N-gly status, proliferation and SHM rates will lead to comprehensive understanding of both N- gly-dependent and independent pro-survival and proliferative transcriptional signatures.
Since the data detailing the differences between FL subclones with different N-gly status and their effects on proliferation is not extensively explored in this review, the section should be modified to describe the actual content of the review. Role of future studies can be mentioned in the appropriate sections. The findings from Haebe et al regarding the gene expression profile can be described in more detail.
Response 1:
‘examines current literature’ part of the abstract has been modified ‘with a particular focus on N-gly-negative subclones’ to avoid confusion – line 34.
More discussion of Haebe results has been added. Lines 182-183, 186-188,
We would like to point out that since our manuscript (Odabashian, Blood, 2020) where we described the first observation of the rare N-gly-negative proliferating FL subclones there were only two publications that picked this discovery up and expanded our knowledge on N-gly-negative subclones’ behaviour (Haebe Blood 2023 and van Bergen Blood Adv 2023). The available literature is limited to these publications so far. This is the point we are making in this perspective – the FL field will benefit with more analysis of bigger multicentred FL cohorts based on the N-gly status and with particular focus on N-gly -negative proliferating subclones as these can instruct us on the major differences between BCR-dependent vs BCR-independent supportive pathways.
Comment 2:
Some of the references did not match the assertions mentioned. For eg –
the statement ‘N-gly negative FL subclones have been observed to upregulate certain metabolic pathways, to enable them to survive and proliferate transiently, despite presumably absent BCR signalling. [11].
Another instance:
In their single cell analysis of 17 longitudinal samples, Haebe et al have shown that N-gly negative subclones can persist and even expand under specific circumstances, particularly when met- abolic pathways such as oxidative phosphorylation and glycolysis are upregulated due to an increase activity of LDHA, COX5A, SLC1A4 genes. This suggests they may have a competitive survival mechanism distinct from N-gly positive subclones [11].
Response 2:
In these two instances, we discussed results from Haebe’s manuscript, which was ref 11 and is now ref 14 in the latest version. In their manuscript, Haebe et al. analysed 17 longitudinal FL samples and compared expression profiles of N-gly positive vs negative FL subclones. They observed the upregulation of metabolic pathways reflected in the upregulation of LDHA, COX5A, and SLC1A4 in negative clones and BCR-related pathways in positive (BTK, CD79B).
Comment 3:
Please ensure all references quoted are accurate.
Response 3:
All were now double checked
Comments on the Quality of English Language
OK
Reviewer 4 Report
Comments and Suggestions for Authors
In this article, the authors propose to uncover some of the mechanisms underlying the clonal evolution of follicular B-cell lymphoma. This article is interesting because it highlights protein-carbohydrate interactions and the importance of N-glycosylation alterations in the pathophysiology of follicular lymphoma, as is likely the case in many other cancers. These protein-oligosaccharide interactions involve various lectin-like molecules, such as DC-SIGN, a molecule discussed in this article. These lectin-oligosaccharide interactions are particularly involved in antitumor immune defense and, more generally, in the tumor microenvironment.
The mechanisms discussed concern mutations allowing "the introduction of N-glycosylation (N-gly) amino acid sequence motifs into the immunoglobulin (IgV) variable region by continuous somatic hypermutation (SHM)". These N-gly motifs, carrying oligomannose motifs capable of interacting with lectin-type molecules, may be involved in the survival of malignant clones. The authors cite numerous clinical observations indicating that the presence of hyper-mannosylated motifs at the BCR level could be involved in the survival of N-gly-positive clones. The situation is more complex at the level of N-gly-negative clones, which would be eliminated except for certain minority clones that would survive by involving mechanisms to be discovered. This suggests that while BCR hyper-mannosylation likely plays an important role in the growth of follicular lymphoma, there are other mechanisms involved in N-gly-negative clones. Indeed, independently with possible new mechanisms specifically involved in N-gly-negative clones survivale, other mechanisms involving various mutations and epigenetic modifications which well documented in the literature, could be involved in the N-gly-negative clones survivale.
On the other hand, can the authors answer these few questions?
1- What is the structure(s) of the hyper-mannosylated oligosaccharides found at the N-gly sequences of the BCR?
2- Is there heterogeneity in the oligosaccharide sequences carried by the N-gly motifs of the BCR of clones within a tumor site?
3- Is there variability in the oligosaccharide motifs of the BCR between different patients?
4- Is there hyper-mannosylation at the level of the N-glycans of other glycoproteins of malignant B lymphocytes?
Finally, to facilitate understanding and strengthen your perspective, it might be helpful to provide readers with:
a) a table outlining other mutations and epigenetic modifications that may play a role in oncogenesis and pathogenicity (e.g., KMT2D, RRAGC, TNFAIP3, TNFRSF14, FOXO1, etc.). In this regard, the results obtained in the EPIC (500,000 individuals) and CPS (120,000 individuals) cohorts show that the association of the t(14-18) translocation with the epigenetic modification CREBBP could be the precursor necessary for cell cancerization (J. Schroers-Martin et al. Tracing Founder Mutations in Circulating and Tissue-Resident Follicular Lymphoma Precursors. Cancer Discov, 2023; doi: 10.1158/2159–8290.CD-23–0111)
b) a simple diagram reminiscent of the lymphoma tumor microenvironment content, with principal cell and molecule interactions (BCR/DC-SIGN…).
Author Response
Dear Reviewers,
We sincerely thank you for your positive and constructive feedback on our manuscript. We are grateful for your suggestions, which have helped us to improve the quality of our manuscript.
Two of the four reviewers noted that the English language could be improved too and that some typos also should be addressed. Please note that the manuscript is written in UK English. In response, we have thoroughly edited the text to enhance readability and have corrected several typos. All revisions and additional text have been highlighted in yellow in the latest version of the manuscript for your convenience.
Please find our detailed responses to each of the reviewers’ comments below.
Reviewer 4
Comments and Suggestions for Authors
In this article, the authors propose to uncover some of the mechanisms underlying the clonal evolution of follicular B-cell lymphoma. This article is interesting because it highlights protein-carbohydrate interactions and the importance of N-glycosylation alterations in the pathophysiology of follicular lymphoma, as is likely the case in many other cancers. These protein-oligosaccharide interactions involve various lectin-like molecules, such as DC-SIGN, a molecule discussed in this article. These lectin-oligosaccharide interactions are particularly involved in antitumor immune defense and, more generally, in the tumor microenvironment.
The mechanisms discussed concern mutations allowing "the introduction of N-glycosylation (N-gly) amino acid sequence motifs into the immunoglobulin (IgV) variable region by continuous somatic hypermutation (SHM)". These N-gly motifs, carrying oligomannose motifs capable of interacting with lectin-type molecules, may be involved in the survival of malignant clones. The authors cite numerous clinical observations indicating that the presence of hyper-mannosylated motifs at the BCR level could be involved in the survival of N-gly-positive clones. The situation is more complex at the level of N-gly-negative clones, which would be eliminated except for certain minority clones that would survive by involving mechanisms to be discovered. This suggests that while BCR hyper-mannosylation likely plays an important role in the growth of follicular lymphoma, there are other mechanisms involved in N-gly-negative clones. Indeed, independently with possible new mechanisms specifically involved in N-gly-negative clones survivale, other mechanisms involving various mutations and epigenetic modifications which well documented in the literature, could be involved in the N-gly-negative clones survivale.
On the other hand, can the authors answer these few questions?
Comment 1:
1- What is the structure(s) of the hyper-mannosylated oligosaccharides found at the N-gly sequences of the BCR?
Response 1:
Added text in lines 58-59
Comment 2:
2- Is there heterogeneity in the oligosaccharide sequences carried by the N-gly motifs of the BCR of clones within a tumor site?
Response 2:
Added discussion in line 60
Comment 3:
3- Is there variability in the oligosaccharide motifs of the BCR between different patients?
Response 3:
We assume the question is about N-gly sequence motif. There is. The N-gly motif sequence N-X-S/T can be lost (e.g. S/T is lost) or changed (e.g. different N-X introduced by SHM – the change of the motif but N-gly remains) by ongoing SHM. The N-gly motif sequence N-X-S/T can vary between patients and within FL in some cases. This fact has been discussed. Added text in lines 120-122 in the text and Figure 1 (an example of the variability within a single patient, Odabashian, Blood, 2020).
Comment 4:
4- Is there hyper-mannosylation at the level of the N-glycans of other glycoproteins of malignant B lymphocytes?
Response 4:
It is a very interesting question and it is possible but outside of this review scope. The main point of this review is the role of the antigen-independent BCR activation through mannose/lectin interaction and how the outcome for FL subclones changes when glycosylation motif and therefore unusual FL IgV N-gly is lost.
Comment 5:
Finally, to facilitate understanding and strengthen your perspective, it might be helpful to provide readers with:
a)a table outlining other mutations and epigenetic modifications that may play a role in oncogenesis and pathogenicity (e.g., KMT2D, RRAGC, TNFAIP3, TNFRSF14, FOXO1, etc.). In this regard, the results obtained in the EPIC (500,000 individuals) and CPS (120,000 individuals) cohorts show that the association of the t(14-18) translocation with the epigenetic modification CREBBP could be the precursor necessary for cell cancerization (J. Schroers-Martin et al. Tracing Founder Mutations in Circulating and Tissue-Resident Follicular Lymphoma Precursors. Cancer Discov, 2023; doi: 10.1158/2159–8290.CD-23–0111)
Response 5:
Added in the text, reference 9, added lines 88 – 92
Comment 6:
b)a simple diagram reminiscent of the lymphoma tumor microenvironment content, with principal cell and molecule interactions (BCR/DC-SIGN…).
Response 6:
Added Figure 3
Reviewer 1
Comment 1:
- Provide a figure/schematic overview showing the interactions between FL cells, BCR, N-glyc and mannose/lectins
Response 1:
Added Figure 3
Comment 2:
- The simple summary should be re-written to a more simple overview
Response 2:
The lay summary has been further simplified
Comment 3:
- First paragraph the % survival rate at 10 years should be added
Response 3:
~80%, Added line 44; added additional ref 3
Comment 4:
- The "certain genetic and epigenetic drives" should be explained in the second paragraph
Response 4:
Added lines 54-55 and more discussion in lines 88 to 92, added additional ref 9
Comment 5:
- Initial drug response rate is mentioned however the standard of care should be explained/outlined here.
Response 5:
Added lines 46 to 50 and ref 4
Comment 6:
- Does N-glycosylation affect outcome to drug treatment in FL? This should be discussed
Response 6:
Virtually all FL cases (apart from rare ones) have N-gly in IgV. N-glycosylation plays a role in the persistence of FL clones by providing survival advantages even in the presence of treatment. This is the main evident effect of IgV glycosylation in FL so far. This has been discussed in the text lines 126-129. Importantly, blocking the mannose/lectin interaction with monoclonal antibodies or other means is under investigation as a potential therapeutic strategy. Lines 136-138 and 261-264.
Comment 7:
- A few typos were noticed throughout
Response 7:
Additional checks have been conducted, the manuscript is written in UK English
Reviewer 2
Comments and Suggestions for Authors
This is a timely review that describes how IgV N-gly motifs are acquired by SHM and its role in BCR activation and the pathogenesis of follicular lymphoma and would be of considerable interest in readers. Most N-gly negative clones are eliminated, but a small percentage can survive and proliferate. Negative clones have been shown to have a gene expression profile similar to light zone cells. The authors recommend pursuing transcriptomic profiling of these different clones to understand the role of N-gly in the context of pathogenic pathways in follicular lymphoma.
Comment 1:
Review is well written and the different sections are sequenced well. However the section describing the aim of the review is confusing. According to the authors, this perspective examines current literature and explores whether a detailed transcriptomic and functional comparison of FL subclones characterised by different N-gly status, proliferation and SHM rates will lead to comprehensive understanding of both N- gly-dependent and independent pro-survival and proliferative transcriptional signatures.
Since the data detailing the differences between FL subclones with different N-gly status and their effects on proliferation is not extensively explored in this review, the section should be modified to describe the actual content of the review. Role of future studies can be mentioned in the appropriate sections. The findings from Haebe et al regarding the gene expression profile can be described in more detail.
Response 1:
‘examines current literature’ part of the abstract has been modified ‘with a particular focus on N-gly-negative subclones’ to avoid confusion – line 34.
More discussion of Haebe results has been added. Lines 182-183, 186-188,
We would like to point out that since our manuscript (Odabashian, Blood, 2020) where we described the first observation of the rare N-gly-negative proliferating FL subclones there were only two publications that picked this discovery up and expanded our knowledge on N-gly-negative subclones’ behaviour (Haebe Blood 2023 and van Bergen Blood Adv 2023). The available literature is limited to these publications so far. This is the point we are making in this perspective – the FL field will benefit with more analysis of bigger multicentred FL cohorts based on the N-gly status and with particular focus on N-gly -negative proliferating subclones as these can instruct us on the major differences between BCR-dependent vs BCR-independent supportive pathways.
Comment 2:
Some of the references did not match the assertions mentioned. For eg –
the statement ‘N-gly negative FL subclones have been observed to upregulate certain metabolic pathways, to enable them to survive and proliferate transiently, despite presumably absent BCR signalling. [11].
Another instance:
In their single cell analysis of 17 longitudinal samples, Haebe et al have shown that N-gly negative subclones can persist and even expand under specific circumstances, particularly when met- abolic pathways such as oxidative phosphorylation and glycolysis are upregulated due to an increase activity of LDHA, COX5A, SLC1A4 genes. This suggests they may have a competitive survival mechanism distinct from N-gly positive subclones [11].
Response 2:
In these two instances, we discussed results from Haebe’s manuscript, which was ref 11 and is now ref 14 in the latest version. In their manuscript, Haebe et al. analysed 17 longitudinal FL samples and compared expression profiles of N-gly positive vs negative FL subclones. They observed the upregulation of metabolic pathways reflected in the upregulation of LDHA, COX5A, and SLC1A4 in negative clones and BCR-related pathways in positive (BTK, CD79B).
Comment 3:
Please ensure all references quoted are accurate.
Response 3:
All were now double checked
Comments on the Quality of English Language
OK
Reviewer 3
Comments and Suggestions for Authors
Comment 1:
- FL is characterized by acquired N-glycosylation, including oligomannose, in the immunoglobulin variable region. Please list its functions. Please explain why it is not present in other indolent B cell tumors. Please also explain why it is present in BL and DLBCL.
Response 1:
We and others have shown the role on the unusual N-gly of FL IgVs in the antigen-independent activation of FL BCR through lectin/mannose interaction that is enough for BCR crosslinking and activation of downstream signalling. Similar motifs can be found in some other B-cell malignancies which has been discussed, please see lines 222-236 In short, the B cell malignancies with a likely antigen dependent activation of BCR (MALT, CLL) very rarely have N-gly motifs in IgV, the same as in healthy B cells. The GC derived lymphomas have higher N-gly prevalence with FL the highest. The differences in the N-gly location (CDR vs FR) in FL vs DLBCL and BL warrant further investigation and currently has no satisfactory explanation (distinct origin, selection pressures, mutation load?).
Comment 2:
- Are there cases without acquired N-glycosylation in the immunoglobulin variable region where there is no N-acetylsylation? Please explain.
Response 2:
As mentioned in the manuscript there are rare clinical cases without N-gly, e.g. in Haebe Blood 2023, one patient had negative FL clones whereas the other three had high prevalence of negative subclones. To reiterate, these are exceptional cases. These unique cases as well as the cases where N-gly-negative subclones are rare but still can be found, provide the opportunity to explore and contrast the BCR-dependent vs BCR-independent survival strategies. The aim of this manuscript is to encourage the FL community to exploit this opportunity.
Comment 3:
- Since N-glyconegative subclones persist temporarily, have alternative survival mechanisms been elucidated? Please explain.
Response 3:
In this manuscript we explain that the FL community is at the early stage of exploring your question. It has been 4 years since these subclones were observed for the first time. Haebe Blood 2023 and van Bergen Blood Adv 2023 are first two follow up studies that highlighted changes in the survival strategy in N-gly-negative subclones with activation of glycolysis, oxidative phosphorylation, fatty acid metabolism, MTORC1 signalling as well as assuming more of LZ B-cell expression profile. This perspective discusses the alternative pathways uncovered by Haebe and van Bergen. We are anxious to see more of these studies and urge the experts in the field to investigate this problem further.
Comment 4:
- Although the initial drug response rate of FL is high, there is a problem with resistance to existing drugs, and transformation to DLBCL is cited as one of the reasons why more than 28% of patients are resistant to treatment. Is N-glycosylation, including oligomannose, in the immunoglobulin variable region enhanced before and after transformation to DLBCL? Please explain.
Response 4:
The current multicentred results imply that N-glycosylation IgV region remains stable before and after transformation from FL to tFL or DLBCL. But more longitudinal paired samples analysis still required for the definite answer to your question.
Round 2
Reviewer 4 Report
Comments and Suggestions for Authors
No further comments. That's fine with me. I found your article very interesting.
Sincerely.
Author Response
Comment: No further comments. That's fine with me. I found your article very interesting.
Response: The authors are grateful for your review of this manuscript, which helped significantly improve it.